# Alternating Updates for Efficient Transformers

**Cenk Baykal**[*]
Google Research

**Dylan Cutler**
Google Research

**Nishanth Dikkala**
Google Research

**Nikhil Ghosh**[†]
UC Berkeley

**Rina Panigrahy**
Google Research

**Xin Wang**
Google Research

## Abstract

It has been well established that increasing scale in deep transformer networks leads to improved quality and performance. However, this increase in scale often comes with prohibitive increases in compute cost and inference latency. We introduce Alternating Updates (AltUp), a simple-to-implement method to increase a model's capacity without the computational burden. AltUp enables the widening of the learned representation, i.e., the token embedding, while only incurring a negligible increase in latency. AltUp achieves this by working on a subblock of the widened representation at each layer and using a predict-and-correct mechanism to update the inactivated blocks. We present extensions of AltUp, such as its applicability to the sequence dimension, and demonstrate how AltUp can be synergistically combined with existing approaches, such as Sparse Mixture-of-Experts models, to obtain efficient models with even higher capacity. Our experiments on benchmark transformer models and language tasks demonstrate the consistent effectiveness of AltUp on a diverse set of scenarios. Notably, on SuperGLUE and SQuAD benchmarks, AltUp enables up to $87\%$ speedup relative to the dense baselines at the same accuracy.

## 1 Introduction

Contemporary machine learning models have been remarkably successful in many domains, ranging from natural language [6, 20] to computer vision [56, 40]. Many of these successes have come in part through sheer scale. A vast amount of empirical studies justify the conventional wisdom that bigger (models and data sets) is better [19, 24]. Accordingly, state-of-the-art Transformer [48] models often contain billions of parameters and are trained for weeks on enormously large data sets using thousands of AI accelerators. Their immense size leads to prohibitive compute and energy costs [36] and prevents their deployment to resource-constrained applications [31].

To alleviate these costs and enable scalability of modern Transformers, a recent line of works have proposed techniques to increase the capacity of models without drastically increasing the computational costs via conditional computation. A notable paradigm is sparsely-activated networks, such as Mixture-of-Experts (MoE) models [11, 59, 35, 1, 28, 43, 45]. The main idea of MoE is to effectively *widen* each network layer by accessing dynamically invoked parameters, i.e., experts, where each expert corresponds to a small subset of disjoint parameters that can be acted on by the input. During training and inference, a given input to the network is routed to a small subset of experts (parameters) to compute the output. As a result, the computation cost remains small relative to the total number of parameters. This scheme enables models with higher capacity with only a relatively small increase in computation.

---

[*]Correspondence to `baykalc@google.com`.

[†]Work done as an intern at Google Research

37th Conference on Neural Information Processing Systems (NeurIPS 2023).

While prior approaches in conditional computation have predominantly focused on the *processing power* of transformers, there is a research gap in efficiently incorporating *widened learned representations*. Recent works have empirically and theoretically established that a wider token representation (i.e., a larger model dimension) helps in learning more complicated functions by enabling more information to be packed in the representation vectors [19, 24, 54]. This phenomenon is also evident in modern architectures of increasing capability. For instance, the representation dimension grows from 512 (small) to 768 (base) and 1024 (large, 3B, and 11B) in T5 models [37], and from 4096 (8B) to 8192 (64B) and 18432 (540B) in PaLM models [6]. A widened representation dimension also significantly improves performance for dual encoder retrieval models [33, 34]. However, naively widening the learned representation requires accordingly increasing the model dimension (see Fig. 1), which quadratically increases the amount of computation in the feedforward computation. In light of the above, a natural question arises: can we leverage the benefit of wider representations without incurring the additional cost of wider transformer layers?

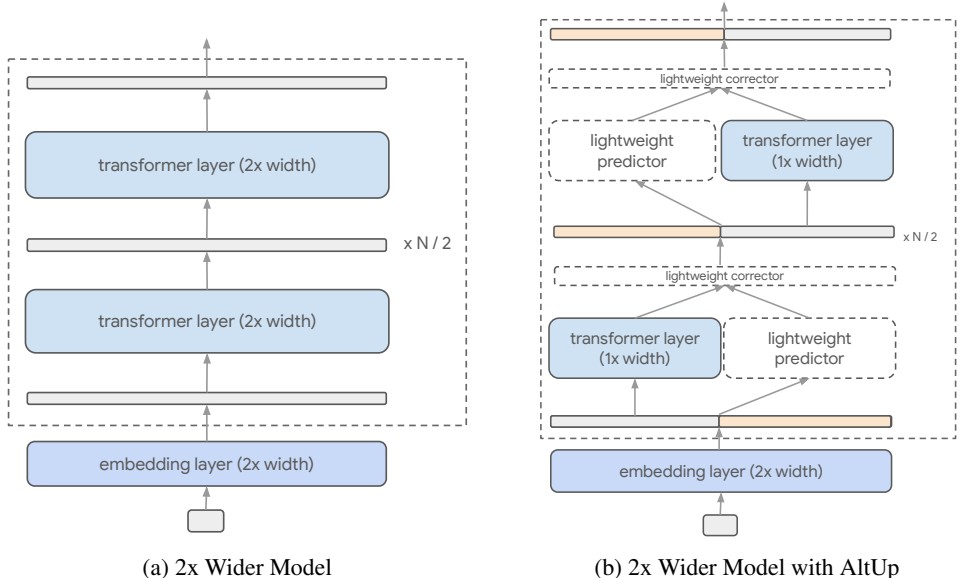

(a) 2x Wider Model           (b) 2x Wider Model with AltUp

Figure 1: An illustration of widening the token representation without (left) and with Alternating Updates (right). This widening causes a near-quadratic increase in computation in a vanilla transformer due to the increased layer width. In contrast, Alternating Updates keeps the layer width constant and efficiently computes the output by operating on a sub-block of the representation at each layer.

In this paper, we address this research gap by introducing *Alternating Updates* (AltUp), a technique to incorporate wider representations in a simple and efficient way. AltUp operates by partitioning the widened representation vector into blocks, processing only a single block at each layer, and using an efficient prediction mechanism to infer the outputs of the other blocks (see Fig. 1). Processing a single block in each transformer layer enables AltUp to simultaneously keep the model dimension, hence the computation cost, constant and take advantage of using an increased token dimension. Unlike prior approaches, e.g., Sparse Mixture of Experts, AltUp is easy to implement, requires minimal hyperparameter tuning, and does not necessitate sharding. Moreover, since AltUp focuses on increasing the representation dimension, it can be applied synergistically with orthogonal techniques like MoE [60] to obtain complementary performance gains.

In particular, our contributions are:

1. We introduce *Alternating Updates* (AltUp) to bridge the research gap in efficiency techniques by enabling wider representations with little additional computation cost. AltUp is simple-to-implement, requires minimal hyperparameter tuning, and does not necessitate sharding.

2. We develop two notable extensions of AltUp: (i) *Recycled-AltUp*, a faster variant of AltUp that requires virtually no additional learnable parameters and (ii) *Sequence-AltUp*, an extension of the AltUp idea to the sequence dimension.

3. We present an extensive evaluation of AltUp on T5 models on various benchmark language tasks. Our experimental results show that AltUp and its variants uniformly lead to models with improved speed-accuracy trade-offs. Notably, on SuperGLUE and SQuAD benchmarks, AltUp enables up to $87\%$ speedup relative to the dense baselines at the same accuracy.

## 2  Related Work

Prior work is rich with a diverse set of techniques to increase the efficiency of contemporary transformer models. Here, we cover the most relevant subset of state-of-the-art techniques and refer the interested reader to [47] for a more comprehensive survey.

Recent works have introduced extremely large, yet scalable models with the use of conditional routing of inputs to a learnable subset of parameters. These sparsely-activated models have achieved state-of-the-art performance on various benchmarks [11] and exhibit favorable theoretical properties [4, 1]. Notably, the Sparse Mixture of Experts (SMoE) [45, 57, 22] family of models use a learned softmax probability distribution to conditionally direct the computation to *experts*, i.e., subsets of network parameters. By routing the computation to a small subset of parameters on an input-dependent basis, SMoE leads to higher capacity models with a relatively small and controllable increase in computation. Switch Transformers [12] show that routing to a single expert on an input-dependent basis reduces computation and outperforms prior SMoE approaches on language tasks. Deja Vu [32] centers around selecting subsets of the attention and MLP parameters to apply for each input (contextual sparsity). Our work is orthogonal to Deja Vu and synergistic with these approaches at large as it focuses on conditional computation.

Follow-up work on SMoE include those that improve the load balancing of experts [60, 29], use reinforcement learning to learn the routing function [7], and leverage smooth top-$k$ expert selection [18] (see [11] for a survey). Other choices for the routing function include non-learnable ones such as Locality Sensitivity Hashing (LSH) [35] which generally maps similar inputs to the same expert, Hash Layers that use token-based hashing [43], and language-specific deterministic routing [10]. Residual Mixture of Experts [51] separates the expert weights into input-independent and input-dependent components.

Conditionally accessing *external memory* is another related approach to vastly increase model capacity at the cost of a relatively small increase in computation [15, 14]. For examples, Memorizing Transformers [53], Memformer [52], and Product key memory [27] leverage dynamic memory to encode and retrieve relevant information. Additional works include those that use an immensely large untrainable corpus, such as Wikipedia, REALM [17], or a 2 trillion token database, RETRO [2]. These prior works that focus on routing(expert)-based mechanisms often necessitate complicated, sharded implementations due to the sheer number of additional parameters that they introduce — often on the order of billions. Our work, on the other hand, is simple-to-implement and requires virtually no hyperparameter tuning. Moreover, AltUp can be synergistically combined with sparsely-activated models like MoE to obtain complementary improvements in efficiency.

Additional relevant works in the realm of efficient transformers include Funnel transformers [8], Reformers [25], Performers [5], Big-Bird [58], and LongT5 [16], among others. These works notably present methods to reduce the quadratic cost of the attention mechanism of transformers. Another flavor of methods complementary to our work is that of adaptive computation, e.g., CALM [44], DynaBERT [21] CascadeBERT [30] and DeeCap [13], where different amounts of computational power is allotted on an example-specific basis via some type of early-exit strategy. AltUp achieves its efficiency via the orthogonal direction of conditionally leveraging wider token representations, and hence can be easily combined with these efficiency techniques and others (e.g., quantization [55], FlashAttention [9]).

## 3  Alternating Updates

In this section, we introduce the method of *Alternating Updates* (AltUp), an approach to enable increased token dimension with little additional computation cost.

## 3.1 Background

At a high level, a standard transformer with $L$ layers generates a $d$-dimensional representation by applying a sequence of layers transformer layers $\mathcal{L}_1, \ldots, \mathcal{L}_L$ as follows. For a particular input token within a sequence of length $N$, the initial token representation $x_1 \in \mathbb{R}^d$ is computed by an embedding table lookup. Subsequently, this $d$-dimensional representation is refined across the transformer layers by iteratively computing $x_{i+1} = \mathcal{L}_i(x_i)$ for each layer $i \in [L]$; here and throughout $[N]$ denotes the set $\{1, \ldots, N\}$ for $N \in \mathbb{N}$. Each transformer layer $\mathcal{L}_i : \mathbb{R}^{d_{\text{model}}} \to \mathbb{R}^{d_{\text{model}}}$ has width $d_{\text{model}}$ (with $d_{\text{model}} = d$ in the standard setting) and contains an attention block and a FeedForward (FFN) block. The width of the layer $d_{\text{model}}$ controls the dimensions of the matrices involved in the attention and FFN blocks. Consequently, the computation cost of attention and FFN scales with $\mathcal{O}(N^2 d_{\text{model}})$ and $\mathcal{O}(N d_{\text{model}}^2)$, respectively. The output, $x_{L+1}$ is the output token representation generated by the transformer. This computation is usually followed by a linear layer operation that maps from the $d$-dimensional representation $x_{L+1}$ to $|\mathcal{V}|$-dimensional logits (in $\mathcal{O}(|\mathcal{V}|d)$ time), followed by a softmax non-linearity to generate the probabilities over the vocabulary $\mathcal{V}$.

Increasing the representation dimension $d$ is a way to enhance the capacity of the transformer model, as a wider representation enables the transformer to store richer information about the input and helps in learning more complicated functions [19, 24, 54]. Naively widening the token representation $d$ requires widening each layer as well, since $d_{\text{model}}$ must match $d$ in a standard transformer model. However, the computation time of each transformer layer grows roughly quadratically with $d_{\text{model}}$, notably for relatively short input sequences. This means that, growing the token dimension from $d$ to $2d$, for example, leads to a model that is at least 2 times (and closer to 4 times for small $N$) slower than the original model with a $d$-dimensional representation.

## 3.2 Alternating Updates

The core idea of Alternating Updates is to *widen the representation vector, but perform computation with a $d$-dimensional sub-block*, and estimate the updated representation using a Predict-Compute-Correct algorithm, as illustrated in Figure 1, right. More specifically, AltUp expands the representation width from $d$ to $Kd$, for integers $K > 1, d > 0$ (for example, $K = 2$ in Fig. 1), but uses layers of width $d_{\text{model}} = d$ (*not* $d_{\text{model}} = Kd$) to transform the representation vector. By keeping the width of each transformer layer constant, AltUp avoids incurring the quadratic increase in computation cost that would otherwise be present with a naive expansion of the representation.

Alg. 1 depicts the details of the per-layer computation involved in a transformer with AltUp with a $Kd$-dimensional representation vector. The input to the AltUp layer is assumed to be the concatenation of $d$-dimensional contiguous subblocks $x_{\text{old}} = \text{concat}(x_{old}^1, x_{old}^2, ..., x_{old}^K) \in \mathbb{R}^{dK}$. Inspired by predictor-corrector methods used to solve ordinary differential equations [3], AltUp first generates a prediction $\hat{x}^i$ for each of the subblocks $i \in [K]$ (Line 1). This prediction takes the form of a mixture of subblocks $\hat{x}^i = \sum_{j=1}^K p_{i,j} x_{old}^j$, where $p_{i,j} \in \mathbb{R}$ for $i, j \in [K]$ are learnable scalars. Subsequently, one of the $K$ sub-blocks is chosen and the computation with the unexpanded transformer layer of width $d_{\text{model}} = d$ is performed on this sub-block (Line 2). Finally, the result of this computation is used in the correction step to generate the updated representation for each sub-block (Line 3).

**Computation time**  We see from Alg. 1 that AltUp introduces negligible amount of additional computation per layer, as the prediction and correction steps involve only vector addition and scalar-vector multiplications ($\mathcal{O}(d)$ operations). Thus, relative to the computation cost of a transformer layer with width $d$ (which we incur on Line 2 in AltUp), the cost of AltUp is only an additional $\mathcal{O}(dK^2)$ per token, where $d$ is the original model dimension and $K$ is the factor of increase in the representation dimension (typically $K = 2$ or $K = 4$, see Sec. 5). This additional $\mathcal{O}(dK^2)$ cost per token is a factor of $d$ smaller than the $\mathcal{O}(d^2 K^2)$ per token cost of the FFN block alone in a $K$-times wider transformer layer. In fact, an AltUp layer does not lead to an increased computation time relative to a $d$-width transformer layer asymptotically, since the $\mathcal{O}(dK^2)$ additional cost per token per layer is dominated by the cost of the FFN block as $K \ll d$ in practice. At a higher level, AltUp requires using an embedding table with width $Kd$ and invoking the final linear operation with $Kd$-dimensional vectors. The initial embedding lookup using a wider table and the linear + softmax operation with $Kd$ (instead of $d$) dimensional vectors may lead to a perceptible increase in computation time. However, since we only incur this additional cost in the beginning and the end, these factors are often inconsequential,

---
**Algorithm 1** Alternating Updates (AltUp) Layer
---

**Input:** $x_{old} = \text{concat}(x_{old}^1, x_{old}^2, ..., x_{old}^K) \in \mathbb{R}^{dK}$: $dK$-dimensional input representation vector to the layer, where $x_{old}^j \in \mathbb{R}^d, j = 1, 2, ..., K$ are contiguous sub-blocks of $x_{old}$.

**Output:** $x_{new} \in \mathbb{R}^{dK}$: The layer's $dK$-dimensional output representation.

1: **Predict**: for each $i \in [K]$, predict the updated representation with a trainable linear map:

$$\hat{x}^i = \sum_{j=1}^{K} p_{i,j} x_{old}^j,$$

where $p_{i,j} \in \mathbb{R}, i, j \in [K]$ are trainable scalars.

2: **Compute**: select a sub-block $j^* \in [K]$ and update this block with $\mathcal{L}$:

$$\tilde{x}^{j^*} = \mathcal{L}(x_{old}^{j^*}).$$

3: **Correct**: for each $i \in [K]$, correct the prediction with the computation result:

$$x_{new}^i = \hat{x}^i + g_i(\tilde{x}^{j^*} - \hat{x}^{j^*}),$$

where $g_i \in \mathbb{R}, i \in [K]$ are trainable scalars.

---

and increasingly so for deeper transformers. Nevertheless, we present an extension to AltUp in Sec. 4 that avoids this slowdown altogether for specialized applications.

**Parameter count**   AltUp introduces $K^2 + K$ additional learnable parameters per layer, where $K^2$ is due to $p_{i,j}, i, j \in [K]$ and $K$ is a result of $g_i, i \in [K]$. Since $K \ll d$, this is an imperceptible amount of additional parameters per layer in practice. Zooming out, AltUp with an expansion factor of $K$ requires a $Kd$-width embedding table, and consequently requires $(K-1)|\mathcal{V}|d$ additional parameters, where $|\mathcal{V}|$ is the vocabulary size. In Sec. 4, we present a variant that requires no additional embedding parameters to be added to the model.

**Memory footprint**   Due to the increase in the representation dimension, AltUp incurs a slightly higher activation memory footprint compared to the baseline model *during training*. In particular, a transformer model with $L$ layers, model dimension $h$, batch size $b$, sequence length $s$, and number of attention heads $a$ has an activation memory footprint of $sbhL(34 + 5as/h)$ [26]. Adding AltUp (with $K = 2$) to this model leads to an additional activation memory of $(sbh + 2sbh)L = 3sbhL$, which is less than 10% of the vanilla transformer's. Moreover, a significant portion of memory is used by the weight parameters which only increase marginally with AltUp. Model parallelism and techniques such as gradient checkpointing, sequence parallelism, or selective activation recomputation can mitigate the memory impact of a wider activation vector even further. During inference, the memory footprint of activations is dominated by the Key-Value cache, which is unaffected by the additional activations introduced by AltUp.

**Selection of sub-blocks**   The selection of the sub-block $j^*$ for the computation step in Algorithm 1 can be any user-specified technique. We consider two simple, deterministic selection methods in this paper and leave more sophisticated methods for future work: (i) **same**: choose the same sub-block for all the layers in a neural network and (ii) **alternating** (default method): for a sequence of layers, alternating through the sub-blocks, that is, if the sub-blocks are indexed with zero-based index, then sub-block $\ell \mod K$ is selected for the computation step for layer $\ell \in [L]$. This alternating selection is the default for Algorithm 1 (hence the name Alternating Updates). We compare the two selection methods empirically in the supplementary material and find that using alternating blocks performs better empirically.

## 4   AltUp Extensions

In this section, we present extensions of the core AltUp idea introduced in the previous section.

## 4.1  Recycled-AltUp: Faster AltUp via embedding recycling

The AltUp formulation presented in Sec. 3 adds an insignificant amount of per-layer computation, however, it does require using a $K$-times wider embedding table. In certain scenarios where the vocabulary $\mathcal{V}$ is very large, this may lead to a non-trivial amount of added computation for the initial embedding lookup and the final linear + softmax operation. A colossal vocabulary may also lead to an undesirable amount of added embedding parameters. *Recycled-AltUp* is an extension of AltUp that avoids these computational and parameter costs by keeping the embedding table's width $d$-dimensional.

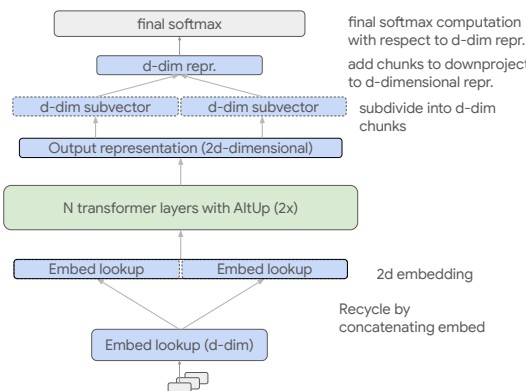

Figure 2: Recycled-AltUp with $K = 2$.

Figure 2 depicts an example application of Recycled AltUp with $K = 2$. The general idea is to *recycle* the initial $d$-dimensional lookup by replicating the $d$-dimensional lookup $K$ times. Hence, Recycled-AltUp virtually adds no additional parameters relative to the baseline width $d$ model. Subsequently, the regular AltUp layers (Alg. 1) are applied until the last linear + softmax operation. To avoid the computational cost of this final operation, Recycled AltUp downprojects the $Kd$-dimensional representation vector $x_{L+1} \in \mathbb{R}^{dK}$ to a $d$-dimensional representation by simply elementwise-adding the $d$-dimensional contiguous sub-blocks in $\mathcal{O}(Kd)$ time. Applying the linear + softmax operation on this down-projected vector implies that the computation cost of this operation is now $\mathcal{O}(|\mathcal{V}|d)$ rather than $\mathcal{O}(K|\mathcal{V}|d)$, effectively reducing the amount of computation by $\mathcal{O}((K-1)|\mathcal{V}|d)$. Our results in Sec. 5 show that Recycle-AltUp's improved speed and reduced parameter count may make it more appealing for certain applications.

## 4.2  Sequence-AltUp: Extension of AltUp to the sequence dimension

Here, we introduce *Sequence-AltUp*, a natural extension of Alternating Updates to reduce the apparent sequence length. This extension is motivated by the computation cost associated with the cost of the attention mechanism for long input sequence lengths. Our approach is similar in its goal to that of prior techniques focused on designing efficient attention mechanisms to reduce the quadratic dependency of attention cost on the sequence length: Funnel transformers [8], Reformers [25], Performers [5], Big-Bird [58], and LongT5 [16].

Similar to the Funnel transformer [8], Sequence-AltUp uses a simple striding operation to reduce the sequence length. Only sampled tokens are processed by the transformer layer while the rest of the tokens require little computation, leading to a computation cost reduction by a factor of $k$, where $k$ is the stride parameter. In this way, Sequence-AltUp is similar to AltUp in that it applies the predict-compute-correct algorithm (Algorithm 1) to the sequence dimension, but it is different from AltUp in that it does not increase the sequence dimension.

Figure 3 depicts the baseline stride-and-skip technique (left) and the proposed Sequence-AltUp method (right). Given an input sequence of vectors $(x_0, x_1, ..., x_{T-1})$ to a transformer layer $\mathcal{L}$, we propose the following extension of Algorithm 1 to reduce the effective sequence length by a factor of $k$. First, we apply a lightweight predictor on the full sequence to obtain a predicted output $\hat{y} = (\hat{y}_0, ..., \hat{y}_{T-1})$. Next, we subsample the input with a fixed stride $k$ and apply $\mathcal{L}$ on the subsampled input and get computed output $\tilde{y} = (\tilde{y}_0, \tilde{y}_k, \tilde{y}_{2k}, ..., \tilde{y}_{\lfloor(T-1)/k\rfloor*k}) = \mathcal{L}(x_0, x_k, ..., x_{\lfloor(T-1)/k\rfloor*k})$. Finally, we use a lightweight corrector to combine $\hat{y}$ and $\tilde{y}$ to form the final output sequence. This design allows unsampled token vectors to obtain contextual information, even though they are not processed by the transformer layer directly—analogous to the inactivated sub-blocks in original AltUp. In contrast, a simple stride-and-skip approach (Figure 3, left) lacks the ability to bring contextual information to the skipped tokens. We present the full algorithm pseudocode and implementation details of Sequence-AltUp in the supplementary material.

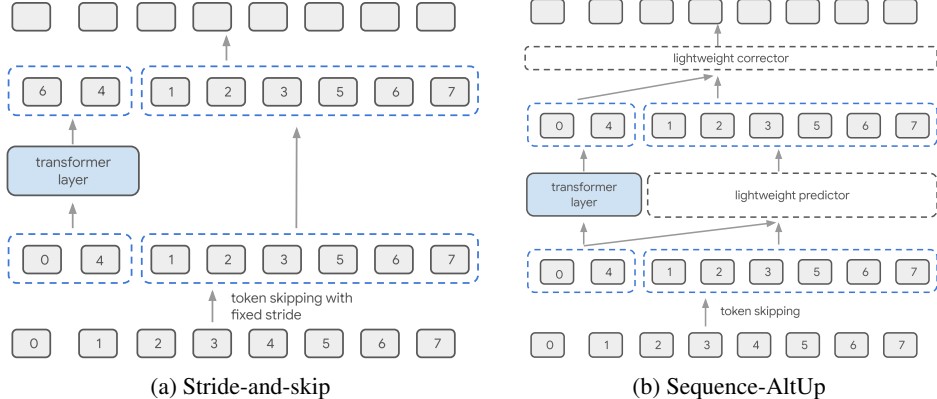

Figure 3: An illustration of Sequence-AltUp (right) and the baseline Stride-and-skip method (left). Sequence-AltUp has virtually the same computation cost as Stride-and-skip, but enables contextual information passing to the skipped tokens.

## 5 Results

In this section, we apply AltUp and its variants to benchmark language models and tasks. We proceed by outlining the experimental setting below. In Secs. 5.1 and 5.2 we present the performance of AltUp on standard benchmarks with varying configurations and model sizes; in Secs. 5.3 and 5.4 we evaluate the performance of AltUp extensions and demonstrate their effectiveness. We present the full details of our evaluations and additional experimental results in the supplementary; namely, the supplementary contains additional evaluations that demonstrate the synergistic combination of AltUp with other conditional compute techniques, additional finetune results, and complementary ablation studies. Overall, our results consistently show that AltUp and its variants enable sizeable performance gains, e.g., up to $87\%$ faster models, across all evaluations on standard benchmarks.

**Setting** We performed all of our experiments using T5-model architectures [37] of varying sizes (small, base, large, and 3B) which we pretrained on the C4 dataset for $500,000$ steps with a batch size of $256$. The pretrained models were then finetuned on either the GLUE [50], SuperGLUE (SG) [49], SQuAD [39] or Trivia-QA (closed-book) [23, 42] benchmark tasks for a further $50,000$ steps with a batch-size of $256$. The pretraining task is to predict corrupted text spans, and the finetuning tasks are re-cast into text generation tasks. We report both pretraining and finetuning metrics: for pretraining, we report span prediction accuracy on a hold-out validation set, and for finetuning, we follow the same recipe as the T5 models, see [37] for more details. The supplementary contains the full details of our evaluations and hyperparameters.

### 5.1 Alternating updates on benchmarks

First, we investigate whether incorporating AltUp on a baseline model leads to an unambiguously better model when we consider the predictive performance and *actual observed latency* (not theoretical FLOPS). To this end, we compare the dense T5-Base/Large/XL models to models augmented with AltUp with $K = 2$ on GLUE, SuperGLUE, SQuAD, and TriviaQA (closed-book) finetuning tasks. Figure 4 plots the performance and normalized speed of the evaluated models.

As the figure depicts, models augmented with AltUp are uniformly faster than the extrapolated dense models at the same accuracy. For example, we observe that a T5 large model augmented with AltUp leads to a $27\%$, $39\%$, $87\%$, and $29\%$ speedup on GLUE, SuperGLUE, SQuAD, and Trivia-QA benchmarks, respectively. Moreover, we see that AltUp's relative performance improves as we apply it to larger models (compare relative speedup of T5 Base + AltUp to that of T5 Large + AltUp). This demonstrates the scalability of AltUp to and its improved performance on even larger models. Overall, **AltUp consistently leads to models with better predictive performance than the corresponding baseline models with the same speed on all model sizes and benchmarks.**

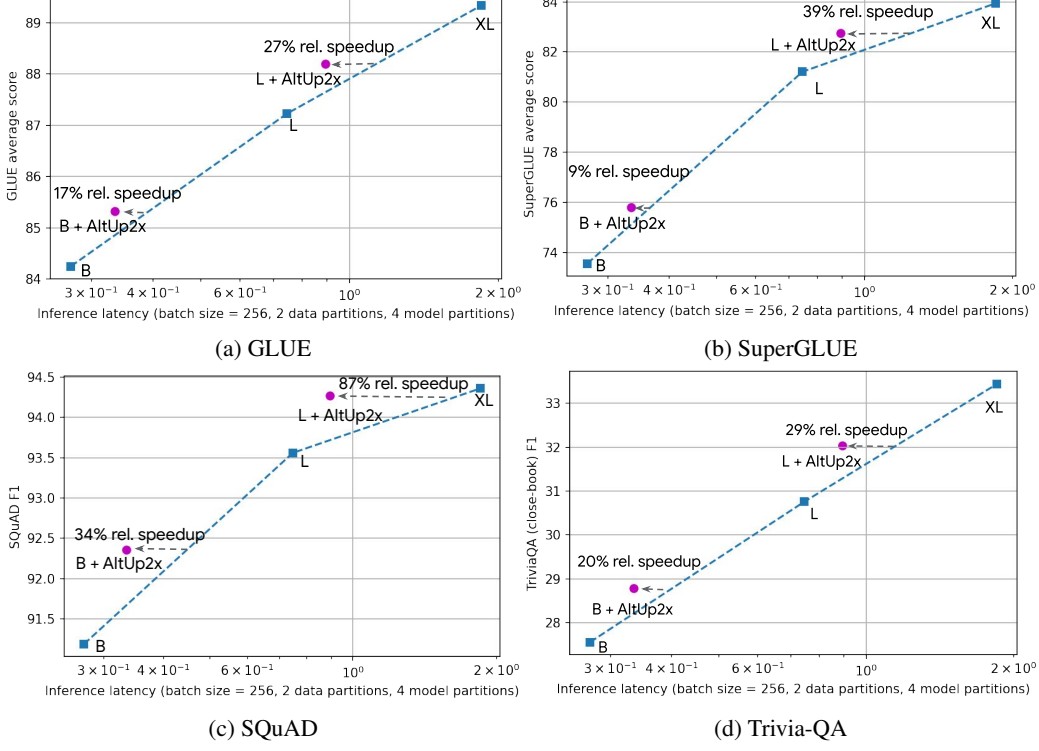

(a) GLUE

(b) SuperGLUE

(c) SQuAD

(d) Trivia-QA

Figure 4: Evaluations of AltUp on T5 models of various sizes and popular benchmarks. AltUp consistently leads to sizeable speedups relative to baselines at the same accuracy. Latency is measured on TPUv3 with 8 cores. Relative speedup is defined as latency delta divided by AltUp latency.

## 5.2 AltUp with varying representation size

In the previous subsection, we had used a value of $K = 2$ for the runs with AltUp. As discussed in Sec. 3, $K$ controls the width of the representation vector, and is the only hyperparameter required by AltUp. Can we obtain even more performant models by using a larger expansion factor $K$? Here, we compare the performance of AltUp with $K = 2$ to AltUp with a larger expansion factor $K = 4$.

Table 1: Performance of AltUp with varying representation dimension scaling parameter $K$ on T5.

| Model | Pretrain Accuracy | GLUE | SG | SQuAD (EM/F1) | TriviaQA (EM/F1) |
|---|---|---|---|---|---|
| S | 61.21 | 75.83 | 59.52 | 76.44/84.97 | 19.03/22.83 |
| S + AltUp (K=2) | 61.86 | **76.82** | **59.60** | **77.51/85.79** | **19.27/22.95** |
| S + AltUp (K=4) | **62.00** | 76.40 | 59.54 | 76.38/84.86 | 19.07/22.84 |
| B | 66.42 | 84.25 | 73.56 | 83.78/91.19 | 23.1/27.56 |
| B + AltUp (K=2) | 66.96 | 85.32 | 75.80 | **85.24/92.36** | 24.35/28.78 |
| B + AltUp (K=4) | **67.18** | 84.95 | **78.91** | 84.82/92.07 | **24.41/28.90** |
| L | 69.13 | 87.23 | 81.21 | 86.77/93.56 | 26.15/30.76 |
| L + AltUp (K=2) | 69.32 | 88.20 | 82.75 | **87.81/94.29** | 27.10/32.04 |
| L + AltUp (K=4) | **69.55** | 88.42 | 82.94 | 87.59/94.02 | **27.36/32.42** |

Table 1 summarizes the results with AltUp instantiated on T5 small, base, and large sized models with hyperparameter $K = 2$ and $K = 4$. We observe that a larger value of $K = 4$ leads to strict improvements in pretrain accuracy over AltUp with $K = 2$ for all models (Table 1, column 2). This is perhaps intuitive, as a wider representation vector enables more information to be learned during the pretraining stage. Interestingly, however, a larger $K$ does not always lead to better finetune performance, especially for smaller models. For example, despite having a worse pretrain accuracy, AltUp with $K = 2$ is better than AltUp with $K = 4$ on all finetune tasks GLUE, SuperGLUE, and SQuAD. We see a similar phenomenon occur for the Base model, but here $K = 4$ is better on GLUE;

and on Large, the trend reverses: $K = 4$ is better on every metric except for SQuAD. Our results indicate that a larger value of $K$ has potential to increase the performance of models on pretrain and fine-tune metrics when AltUp is applied to larger models. *We note that there is an inherent trade-off between a larger factor $K$ and trainability, however, as a larger value of $K$ leads to less frequent activation of each sub-block which may impair performance.* We envision that practitioners can pick a value of $K$ other than the default $K = 2$ to optimize performance on an application-specific basis.

## 5.3 Recycled-AltUp

Next, we consider the performance of the lightweight extension of AltUp, Recycled-AltUp, introduced in Sec. 4. We apply Recycled-AltUp with $K = 2$ to T5 base, large, and XL models and compare its pretrain accuracy and speed to those of baselines. We record both the training speed and inference speed of the resulting models. Since Recycled-AltUp does not require an expansion in the embedding table dimension (see Sec. 4), we remark that the models augmented with it have virtually the same number of trainable parameters as the baseline models.

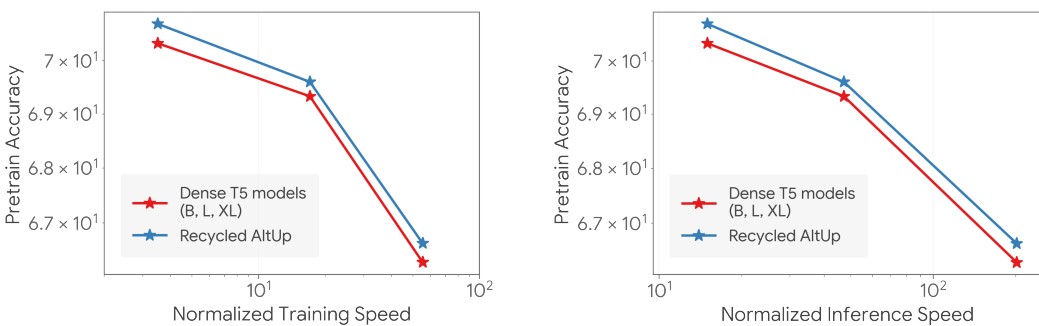

Figure 5: Recycled-AltUp on T5-B/L/XL compared to baselines. Recycled-AltUp leads to strict improvements in pretrain performance without incurring any perceptible slowdown.

The results of our experiments are shown in Fig. 5. The figures for both the training and inference speed show that models with Recycled-AltUp clearly improve over baselines in pretrain accuracy, without any perceptible slowdown. While Recycled-AltUp's predictive strength generally falls below that of standard AltUp (cf., pretrain values for AltUp in Table 1), its improved speed and reduced parameter count may make it more suitable for certain applications – see Recycled-AltUp Fine-tune evaluations in the Appendix (Sec. G). We present additional fine-tuning results with Recycled-AltUp in the supplementary material; overall, our results demonstrate that Recycled-AltUp is similarly effective on fine-tuning tasks.

## 5.4 Sequence-AltUp

Here, we evaluate Sequence-AltUp (from Sec. 4) to reduce the apparent sequence length for the T5 base model. In particular, we apply average pooling, stride-and-skip, and Sequence-AltUp to the encoder layers to reduce the apparent input sequence length. We apply stride-and-skip and Sequence-AltUp to layers $2, \ldots, L - 1$ of the encoder, rather than all the layers, with stride length $4$ as we found that this configuration results in a better accuracy/speed trade-off for both techniques. For average pooling, the sequence length is immutably reduced from the start according to the method.

Table 2: Performance and pretrain speed of different methods for sequence length reduction on T5.

| Model | Pretrain Accuracy | Finetune GLUE | Finetune SG | Speed |
|---|---|---|---|---|
| S | 61.21 | 59.52 | 76.44/84.97 | 166.1 |
| B (Baseline) | 66.42 | 73.56 | 83.78/91.19 | 52.4 |
| Average pooling | 63.89 | 57.85 | 71.37/81.87 | 91.9 |
| Stride-and-Skip | 65.02 | 65.98 | 79.72/87.64 | 79.4 |
| Sequence-AltUp | **65.39** | **66.94** | **81.67/89.37** | 74.9 |

Table 2 presents the comparisons on pretrain and finetune metrics (GLUE and SuperGLUE) and pretrain speed (measured by the number of sequences per second per core). The table additionally lists the relevant metrics for T5 Base (which is the baseline model) and T5 Small as reference points in the table. We observe that average pooling gives a large speed-up, but suffers from severe quality degradation, especially on the finetune metrics where it performs even worse than T5 small. Stride-and-skip and Sequence-AltUp, on the other hand, offer an improved quality and speed trade-off relative to T5 Base. In particular, Sequence-AltUp is only slightly slower than stride-and-skip (yet, still $\approx 40\%$ faster than the baseline), but is much closer to the baseline model's quality.

## 6 Conclusion

We propose the method of *Alternating Updates* (AltUp) to increase the capacity of modern transformer models without incurring a significant increase in latency. Our approach bridges the research gap in efficient transformers by enabling the use of wider token representations without widening the transformer layers. AltUp utilizes lightweight prediction and correction steps to update a wider representation vector without increasing the transformer layer's computation cost. As a result, we achieve strong performance improvements on language modeling and language understanding benchmarks. We present extensions of AltUp that enable additional gains in efficiency. Given its orthogonal scope, AltUp can be synergistically applied with existing techniques like MoE. On popular language understanding and QA benchmarks, AltUp enables up to $87\%$ speedup relative to the dense baselines at the same accuracy.

**Limitations and future work**  A current limitation of the technique we propose is the lack of a deep theoretical understanding of its properties due to the complicated nature of rigorously analyzing transformer models. An interesting open question is whether it would be possible to analyze AltUp by relating its performance to a block compressed layer, and transitively relating that to a wide layer without block compression. A deeper understanding of AltUp may also shed light on the optimal hyperparameter $K$ on an application-specific basis. In future work, we plan to conduct a theoretical investigation of alternating updates to develop a deeper understanding of its effectiveness across differing applications. We also plan to experiment with the use of a very large expansion factor $K$.

**Broader Impact**  Training and deploying modern neural network models consumes colossal amounts of resources. This leads to detrimental effects on the environment and hampers the widespread applicability and democratization of AI. We envision that AltUp can serve as a valuable component of efficient architectures of the future and help alleviate these negative impacts.

## Acknowledgments

We would like to thank Ravi Kumar, Erik Vee, and Nikhil Vyas for constructive discussions and their feedback.

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

# Supplementary Material for *Alternating Updates for Efficient Transformers*

In this supplementary, we present the full details and hyperparameters of our evaluations, provide details of our algorithmic contributions, and present complementary empirical results that support the effectiveness the presented work.

## A    Experimental Setup

We experiment with various T5 baseline model sizes: small (S), base (B), large (L), and XL. The base (B), large (L), and XL models follow the same model configurations as in the T5 paper, while the small model is shallower than the T5 paper [37] to cover a larger range of model sizes (4 encoder/decoder layers instead of 8 encoder/decoder layers). In particular, we use the T5 version 1.1 models with gated GELU feedforward network and pre layer norm. The models are implemented on top of the T5X [41] codebase. During pretraining, we use 256 batch size, Adafactor optimizer [46] with base learning rate 1.0 and reciprocal square-root decay with 10000 warmup steps, and zero dropout. During finetuning, we use 256 batch size, Adafactor optimizer with constant learning rate of 0.001 and 0.1 dropout. Unless explicitly stated, we pretrain for $500k$ steps and finetune for $50k$ steps. Our experiments were implemented in Python and run on TPUv3 with 8 cores.

## B    Parameter Counts and Speed

Here, we present the number of additional parameters needed by adding AltUp, its speed, and its pretrain accuracy on T5 models of varying sizes. In the following tables, the embedding parameters include input embedding table parameters (shared between encoder and decoder) and output embedding table. Non-embedding parameters include all the transformer blocks. Train speed is measured by number of examples per second per core. Table 3 documents the parameter count and training speed comparison. Note that Alternating Updates increases the number of embedding parameters while leaving the non-embedding parameters roughly the same. Since the narrow transformer layer computation is not changed by alternating updates and since the predict and correct steps are lightweight (see Sec. 3), we incur a relatively small increase in the computation cost compared to a dense 2x width model.

| Model | # emb params | # non-emb params | train speed |
|---|---|---|---|
| S | 3.29E+07 | 3.78E+07 | 166.1 |
| S + AltUp | 6.58E+07 | 3.99E+07 | 119.4 |
| B | 4.93E+07 | 1.98E+08 | 52.4 |
| B + AltUp | 9.87E+07 | 2.12E+08 | 42.3 |
| L | 6.58E+07 | 7.17E+08 | 17.1 |
| L + AltUp | 1.32E+08 | 7.68E+08 | 14.4 |

Table 3: Model size and train speed comparisons on T5X models with AltUp instantiated with $K = 2$.

Table 4 documents the parameter count, training speed and pretrain accuracy comparison when the representation dimension is scaled up with AltUp or dense scaling. Note that Alternating Updates increases the number of embedding parameters while leaving the non-embedding parameters roughly the same, providing an efficient way to scale up the representation dimension relative to a $K$-times wider model.

Table 5 contains pretrain performances for T5 XL sized models. We note the AltUp technique continue to offer quality boost at the billion parameters scale (note that T5XL has roughly 3B parameters), suggesting that AltUp is a robust technique for increasing model capacity for modern large language models.

## C    Combination with MoE

Here, we investigate whether AltUp can be combined with orthogonal techniques, namely MoE, to obtain additive performance gains in pretrain accuracy for T5 small, base, and large models. In

| Model | # emb params | # non-emb params | Train speed | Pretrain accuracy |
|---|---|---|---|---|
| T5 Base | 4.93E+07 | 1.98E+08 | 52.4 | 65.29 |
| T5 Base + AltUp2x | 9.87E+07 | 2.12E+08 | 42.3 | 65.78 |
| T5 Base + Dense2X | 9.87E+07 | 3.97E+08 | 32.9 | 66.45 |
| T5 Base + AltUp4x | 1.97E+08 | 2.41E+08 | 28.1 | 66.00 |
| T5 Base + Dense4X | 1.97E+08 | 7.93E+08 | 12.6 | 67.01 |

Table 4: AltUp compared with dense scaling evaluated at $250k$ pretrain steps.

| Model | # emb params | # non-emb params | Train speed | Pretrain accuracy |
|---|---|---|---|---|
| T5 XL | 1.32E+08 | 2.72E+09 | 3.6 | 70.01 |
| T5 XL + AltUp2x | 2.63E+08 | 2.92E+09 | 3.0 | 70.61 |

Table 5: Pretrain performances for T5 XL sized models. Pretrain accuracy is measured at $400k$ steps. AltUp continues to offer a performance boost even on the scale of models with billions of parameters.

particular, we consider the *partial experts* setting similar to [38, 35], where at each layer, in addition to the layer's module, we route the input to a smaller expert module and combine the outputs of the main and auxiliary modules as the input to the subsequent layer (see Fig. 6).

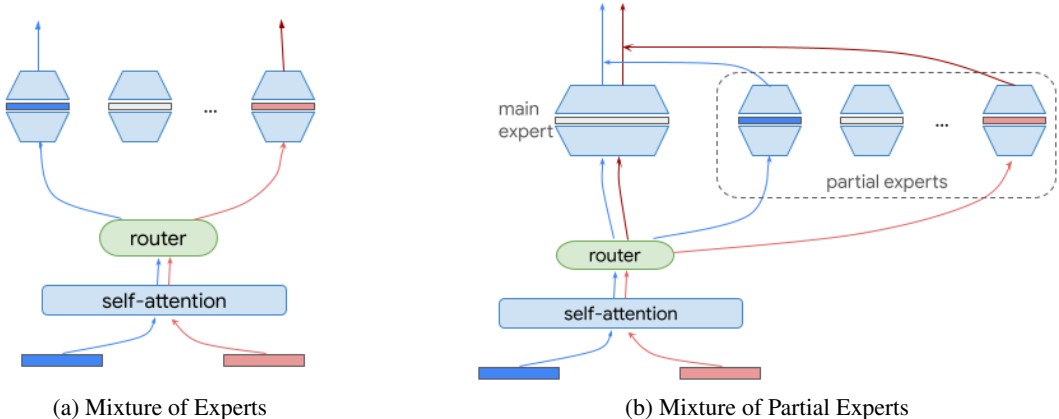

(a) Mixture of Experts

(b) Mixture of Partial Experts

Figure 6: The partial experts setting in the context of the evaluations in Sec. C. The standard MoE model (left) routes the inputs to one or more of $n$ experts based on a routing function. Mixture of Partial Experts (right) always routes the input to the main expert and additionally routes the input to one or more partial experts; the output is a function of the main expert's and partial experts' outputs.

The MoE layer routes an input token $x$ to $k$ of $n$ experts where each expert is itself a parametrized subnetwork (e.g., a fully-connected layer). Following [11], we let $\{E_i(\cdot)\}_{i \in [n]}$ and $E_i(x)$ denote the set of experts and the output of lookup the input token $x$ to expert $i$, respectively. For an input token $x$, a learnable weight matrix $W$ is applied to obtain the logits $h(x) = Wx$. The lookup probabilities are computed by taking the softmax of $h(x)$

$$p_i(x) = \frac{\exp(h_i(x))}{\sum_{j \in [n]} \exp(h_j(x))} \quad \forall i \in [n].$$

The token $x$ is routed to the expert(s) $\mathcal{T} \subset [n]$ with the top-$k$ probabilities $p(x)$. Since this operation is not differentiable, the output $y$ is computed as a probability weighted combination of the experts' outputs to enable gradients to propagate back to the router parameters [45], i.e.,

$$y = \sum_{i \in \mathcal{T}} p_i(x) E_i(x).$$

For MoE, we used the simplified implementation of the top-1 softmax routing of [12]. We use 128 experts each per encoder and decoder layer, with each expert representing a 2 layer fully-connected neural network with hidden dimension 16. For sake of the synergistic demonstration even with the core MoE implementation, we did not incorporate a sophisticated mechanism for load balancing such as load balancing loss [12] or router z loss [60]. We use multiplicative jitter noise sampled from a uniform distribution over $[1 - \varepsilon, 1 + \varepsilon]^{d_{\mathrm{in}}}$ with $\varepsilon = 0.01$. The router matrix $W$ was initialized by drawing from a zero mean Normal distribution with standard deviation $2 \times 10^{-2}$.

| Method | T5 Small | T5 Base | T5 Large |
|---|---|---|---|
| Baseline | 59.10 | 63.35 | 65.58 |
| MoE [60] | 59.42 | 63.62 | 65.71 |
| AltUp (K=2) | 59.67 | 63.97 | 65.73 |
| AltUp (K=2) + MoE | **59.91** | **64.13** | **65.95** |

Table 6: Pretrain accuracy at 100k steps of T5 models augmented with Alternating Updates (see Sec. 3) and MoE. MoE synergistically combines with Alternating Updates and enables further increases in model capacity.

Table 6 synthesizes the pretraining performance on the C4 dataset at 100k training steps of AltUp and compared techniques on T5 Small, Base, and Large models. Perhaps most notably, we show that combining AltUp and MoE leads to even further sizable improvements in the pretrain performance (last row of Table 6). For example, the combination of MoE and AltUp improves over the baseline by 0.81, over AltUp alone by 0.24, and over MoE alone by 0.49. For all model sizes, the combination of AltUp and MoE is synergistic and leads to significant improvements compared to not only the baseline, but also to each approach in isolation.

## D  Alternating updates with varying block selection

In this section, we present empirical results on the alternating updates technique and comparison with other techniques that widen token representation vectors. We increase the token representation dimension by a factor of 2 (corresponding to $K = 2$ in Algorithm 1) unless otherwise specified. The model capacity increase comes from a wider embedding table at the bottom layer of the model while the transformer layers remain the same, which results in minimal additional computation cost.

| Model | Pretrain Accuracy | Finetune GLUE | Finetune SG | Finetune SQuAD (EM/F1) |
|---|---|---|---|---|
| S (baseline) | 61.21 | 75.83 | 59.52 | 76.44/84.97 |
| S + Sum | 61.67 | 77.54 | 59.63 | 75.06/83.82 |
| S + SameUp | **61.91** | **77.75** | **60.81** | 76.85/85.51 |
| S + AltUp | 61.86 | 76.82 | 59.60 | **77.51/85.79** |
| B (baseline) | 66.42 | 84.25 | 73.56 | 83.78/91.19 |
| B + Sum | 66.82 | 84.85 | 75.2 | 84.36/91.36 |
| B + SameUp | 66.82 | 84.06 | 74.15 | 84.41/91.76 |
| B + AltUp | **66.96** | **85.32** | **75.80** | **85.24/92.36** |
| L (baseline) | 69.13 | 87.23 | 81.21 | 86.77/93.56 |
| L + Sum | 69.09 | 86.18 | 78.93 | 86.19/93.08 |
| L + SameUp | **69.45** | 87.95 | 82.72 | **87.65**/94.13 |
| L + AltUp | 69.32 | **88.20** | **82.75** | 87.58/**94.27** |

Table 7: Comparison of Algorithm 1 with various sub-block selection methods on T5-S/B/L.

In Table 7, we compare the summation method (Sum) in which additional embedding vectors are added to the token representation vector, Algorithm 1 with same block selection (SameUp), and Algorithm 1 with alternating block selection (AltUp), all on top of the T5 version 1.1 small (S), base (B), and large (L) models. We observe the Prediction-Compute-Correct scheme as described in Sec. 3 with *same* and *alternating* block selection methods outperforms the summation method. For the small models, *same* block selection method performs better in most tasks, while for the base and

large models, *alternating* block selection method performs better in most tasks. We note all three methods bring improvements in both pretraining and finetuning, and AltUp is generally the most effective one. While pretraining accuracies for all three methods are mostly similar, differences in finetuning metrics are large, and AltUp generally achieves roughly twice the gains of the other two variants. Moreover, we observe that the gains of AltUp in pretraining accuracies show diminishing returns when model sizes grows, but the gains in finetuning metrics do not.

## E  Additional Evaluations

In addition to the T5-based evaluations presented in the main body of the paper, we conducted a preliminary study on a lightweight BERT model. The model has 12 layers, 256 model dimension, and 4 attention heads. On a masked language pretraining task (with BERT pretraining data), lightweight-BERT achieves $54.7$ MLM accuracy while lightweight-BERT + AltUp ($K = 2$) achieves $56.2$ MLM accuracy.

In Fig. 4, we observed that SuperGLUE is particularly exceptional in terms of the diminishing returns with respect to the model capacity. For example, in any of the other plots (GLUE, SQuAD, Trivia-QA), we see that the dense baselines fall well below the lines formed by AltUp's data points. In order to obtain a more direct comparison on SuperGLUE, we compared L + AltUp with a size-matched dense baseline – T5L+4, which adds 4 encoder layers and 4 decoder layers to the T5L model. Note this model has the same number of parameters as L + AltUp, but uses more computation and is roughly $10\%$ slower than L + AltUp. On SuperGLUE, T5L+4 achieves an average score of $82.57$, versus $82.75$ for L + AltUp. Therefore, AltUp is not only faster, but more performant when compared to a dense baseline. Note that this is specific to SuperGLUE, and we see larger speedups – up to $87\%$ – on other datasets.

## F  Sequence-AltUp Details

Here, we provide the full pseudocode of Sequence-AltUp from Sec. 4 (see Alg. 2).

---

**Algorithm 2** AltUp extension to sequence dimension

---

**Input:** A sequence of vectors $x = (x_0, x_2, ..., x_{T-1})$, where $x_i \in \mathbb{R}^d$. Transformer layer $\mathcal{L}$ and stride parameter $k$.

**Output:** A sequence of vectors $y = (y_0, y_2, ..., y_{T-1})$.

1: **Prediction**: predict the output sequence with a trainable linear map:

$$\hat{y}_i = a_1 x_i + a_2 x_{\lfloor i/k \rfloor * k}$$

for $i = 0, 1, ..., T - 1$, where $a_1, a_2 \in \mathbb{R}$ are trainable scalars;

2: **Computation**: subsample the input sequence with stride $k$ and apply the transformer layer on the subsampled sequence:

$$(\tilde{y}_0, \tilde{y}_k, ..., \tilde{y}_{\lfloor (T-1)/k \rfloor * k}) = \mathcal{L}(x_0, x_k, ..., x_{\lfloor (T-1)/k \rfloor * k});$$

3: **Correction**: correct the prediction with the computation result:

$$y_i = \hat{y}_i + b(\tilde{y}_{\lfloor i/k \rfloor * k} - \hat{y}_{\lfloor i/k \rfloor * k})$$

for $i = 0, 1, ..., T - 1$, where $b \in \mathbb{R}$ is a trainable scalar.

---

## G  Recycled-AltUp Fine-tune Evaluations

We conclude the supplementary material by presenting evaluations with Recycled-AltUp as described in Sec. 4. Table 8 presents the results of our pretrain and fine-tune evaluations on T5 Small, Base, and Large. The pretrain accuracy is the one reported at $500k$ steps, and we fine-tune for an additional $50k$ steps for the fine-tune evaluations.

Consistent with our results presented in the main body of our paper, we observe that AltUp and Recycled-AltUp both provide clear and consistent gains over the baseline on virtually all pretrain

| Model | Pretrain Acc. | GLUE | SG | SQuAD (EM/F1) | TriviaQA (EM/F1) |
|---|---|---|---|---|---|
| S | 61.21 | 75.83 | 59.52 | 76.44/84.97 | 19.03/22.83 |
| S + Recycled-AltUp | 61.33 | **77.24** | 59.12 | **77.76**/85.64 | 19.06/22.77 |
| S + AltUp | **61.86** | 76.82 | **59.60** | 77.51/**85.79** | **19.27**/**22.95** |
| B | 66.42 | 84.25 | 73.56 | 83.78/91.19 | 23.1/27.56 |
| B + Recycled-AltUp | 66.63 | **85.60** | 74.83 | 84.81/91.93 | 22.72/27.15 |
| B + AltUp | **66.96** | 85.32 | **75.80** | **85.24**/**92.36** | **24.35**/**28.78** |
| L | 69.13 | 87.23 | 81.21 | 86.77/93.56 | 26.15/30.76 |
| L + Recycled-AltUp | 69.30 | 87.91 | 82.53 | 87.37/93.88 | **27.51**/**32.38** |
| L + AltUp | **69.32** | **88.20** | **82.75** | **87.81 / 94.29** | 27.10/32.04 |

Table 8: The performance of baseline, Recycled-AltUp, and AltUp on pretrain and fine-tune evaluation metrics. Recycled-AltUp and AltUp were instantiated with $K = 2$ for all evaluations.

and fine-tune metrics. As conjectured in Sec. 4 Recycled-AltUp generally does not provide the full benefits of AltUp in terms of pretrain and fine-tune accuracies, however, this gap seems to shrink for larger models. Moreover, Recycled-AltUp has the appeal that it practically adds no additional parameters to the model and, as a result, has roughly the same speed as the baseline model (see Fig. 5). Given the lightweight and latency-matching nature of Recycled-AltUp, these improvements directly translate to clear gains over the dense baselines. We envision that Recycled-AltUp's improved speed and reduced parameter count may make it more appealing for certain applications.

