# OpenReview forum: "Alternating Updates for Efficient Transformers"
_NeurIPS.cc/2023/Conference — NeurIPS 2023 spotlight_

### Official Review · Reviewer_2iUa · 2023-06-10

**Soundness:** 3 good
**Presentation:** 3 good
**Contribution:** 2 fair
**Rating:** 6
**Confidence:** 4

**Summary:**

This work proposes an efficient way to increase the width of transformer models, i.e. Alternating Updates (AltUp). AltUp can increase the width of one existing model with little computation overhead. Authors evaluated their approach on T5 model and see some improvements on well-established benchmarks including GLUE SG SQuAD and TriviaQA.

**Strengths:**

1) This paper is well-motivated. Scaling efficiently is indeed an important topic. Considering scaling is so useful, how to save the cost of scaling up is very helpful to our community.
2) Authors conduct a comprehensive evaluation and explored various modifications of this approach, such as Recycled-AltUp and Sequence-Altup.

**Weaknesses:**

My main concern is about the effectiveness of the proposed model. In Figure 4, we can see AltUp cannot bring a very significant improvement. For instance, in Figure 4b, if we connect B+AltUp and L+AltUp, the L without AltUp will almost appear on the line. Certainly, the x-axis is at log scale, it is not totally fair by using this line. However, it shows that the improvement of AltUp is indeed not very significant.

**Questions:**

1) Would AltUp introduce more activation? If yes, although AltUp introduces little flops, given the fixed hardware, the max batch size would be smaller. For instance, if the original T5 base can use batch size = 256 on 16 TPUs, your AltUp would only be able to use batch size = 128. Then, to support a larger global batch size, you have to use gradient accumulation. This issue would be more serious for a longer sequence.

**Limitations:**

Theoretical analysis is good. But as stated in the weakness section,  my main concern is whether the AltUp is working well enough to let more people try and use it. If AltUp is not working well enough on the official paper, since trying a new approach is expensive in scaling transformer, people will not really use the model and then this work would have little impact.

---

> ### Author Rebuttal · Authors · 2023-08-09
>
> Thank you for your careful consideration of our paper and constructive feedback.
>
> ### Effectiveness of AltUp
>
> We would like to point out that AltUp achieves significant speedups in *wall-clock time* (not only theoretical FLOPS) compared to dense baselines as shown in our evaluations in Sec. 5. In fact, AltUp enables up to 87% speedup relative to the dense baselines at the same accuracy for SQuAD and is highly effective on the other evaluated datasets as well.
>
> Regarding your point about Figure 4b: here, we show the comparison of T5 dense baselines and compare them to T5 Base + AltUp2x and T5 Large + AltUp2x. We use a log-scale because it is standard in prior work in this area, e.g., in scaling laws for neural language models [1,2]. As we can see in Figure 4b, there is a clear trend of diminishing returns (even in the log-scale, even for the baselines) as we go from T5L to T5XL. This means that it is not entirely fair to connect the B+AltUp point to L+AltUp and state that L lies on this line, since L+AltUp is well past the point of diminishing returns that the models experience as we increase their capacity. We would like to highlight that SuperGLUE is particularly exceptional in terms of this diminishing returns phenomenon. For example, if we conduct the same point connecting procedure in any of the other plots (GLUE, SQuAD, Trivia-QA), we see that the dense baselines fall well below the lines formed by AltUp’s data points.
>
> In order to make this comparison for SuperGLUE more direct, we compared L + AltUp with a size-matched dense baseline – T5L+4, which adds 4 encoder layers and 4 decoder layers to the T5L model. Note that this model has the same number of parameters as L + AltUp, but uses more computation and is roughly 10% slower than L + AltUp. On SuperGLUE, T5L+4 achieves an average score of 82.57, versus the average score of 82.75 for L + AltUp. Therefore, AltUp achieves better quality even when compared to a compute-heavier dense baseline. Note that this is specific to SuperGLUE, and we see larger speedups – up to 87% – on other datasets. We would be happy to include this discussion in our final submission.
>
> Moreover, we would like to highlight that the lightweight version of AltUp, Recycled-AltUp (from Sec. 4), adds virtually no additional latency or additional parameters to the dense baseline models and strictly improves their performance (see Fig. 5). In Table 8 of the appendix, we also demonstrate that Recycled-AltUp leads to similar gains as AltUp does on GLUE, SuperGLUE, SQuAD, and TriviaQA. Given the lightweight and latency-matching nature of Recycled-AltUp, these improvements directly translate to clear gains over the dense baselines. We will clarify this point in our revision and highlight the effectiveness and practicality of AltUp.
>
>
>
> ### AltUp and Activation Memory Footprint
>
> Although AltUp has a higher activation memory footprint compared to the baseline *during training*, this memory footprint is largely negligible when compared to the activation memory footprint of the baseline. Thus, AltUp would not need a drastic reduction in batch size in practical applications.
>
> Following the computations in “Reducing Activation Recomputation in Large Transformer Models” [1], a transformer model with $L$ layers, model dimension $h$, batch size $b$, sequence length $s$ and number of attention heads $a$ has total activation memory of
>
> $ s b h L (34 + 5 a s / h) $.
>
> When we add AltUp (with $K=2$) on this model, the additional activation memory due to is
>
> $ (s b h + 2 s b h) L = 3 s b h L$,
>
> which is less than 10% of the vanilla transformer’s activation memory footprint. Moreover, a significant portion of memory is used by the weight parameters which only increase marginally with AltUp. Overall this results in a <10% additional memory usage with AltUp. In addition, since the additional blocks are inexpensive to recompute, we can also recompute these blocks in the backprop without storing them. Model parallelism and techniques such as gradient checkpointing, sequence parallelism, or selective activation recomputation can mitigate the impact of a wider activation vector even further.
>
>
> During inference, the memory footprint of activations is mostly due to the size of the KV cache. In AltUp, since each transformer layer takes a single sub-block, the additional activations introduced by AltUp do not show up in the KV cache.
>
> [1] https://arxiv.org/abs/2203.15556
>
> [2] https://arxiv.org/pdf/2001.08361.pdf

---

> > ### Comment · Reviewer_2iUa · 2023-08-10
> >
> > Thank you for your reply. My concerns are solved. I decide to raise my Rating to 6.

---

> > > ### Author Response · Authors · 2023-08-10
> > > **Thank you**
> > >
> > > We are happy to hear that we were able to address your concerns in our rebuttal!
> > >
> > > It seems that the score is not updated in the original review. Could you please update it at your convenience?
> > >
> > > Thank you again for your time and consideration of our paper.

---

### Official Review · Reviewer_raib · 2023-07-02

**Soundness:** 3 good
**Presentation:** 3 good
**Contribution:** 4 excellent
**Rating:** 8
**Confidence:** 4

**Summary:**

The paper proposes a novel technique named “AltUp” to expand Transformer’s feature dimension while preserving the computation cost. The key idea of AltUp is to divide wide hidden features into multiple blocks, where only one block is processed by Transformer sub-layers, while the other blocks are computed through a linear combination of all blocks. The position of the updated block alters across layers. Furthermore, the authors present two variants of AltUp, “Recycled-AltUp” and “Sequence-AltUp”, which focus on reducing the embedding parameters and effective sequence length, respectively. Experimental results show that the models trained with AltUp are up to 87% faster in inference compared to the model with the same performance.

**Strengths:**

* The concept of “widening representations” is unique and hasn’t been explored much before. This approach effectively disentangles the computation and feature dimensions. Importantly, the paper introduces a non-trivial solution that minimizes additional computation costs.
* AltUp demonstrates promising speedup across various model sizes, with K=2 generally performing better than the original version.


**Weaknesses:**

* Experiments are conducted on T5 models; however, the paper could be strengthened if encoder-only or decoder-only architectures are also evaluated.
* The concept of “Sequence-AltUp” may be considered somewhat different from the “widening representations”. Maybe this could be justified by “effective width for each token” … More clarification would be helpful.


**Questions:**

* (Minor) Is Conformer (L214) an appropriate example for “striding operation”?
* (Suggestion) I basically agree with that ‘large K leads to less frequent activation’ (L279-280), but could this issue be compensated by 2-4x longer training steps?


**Limitations:**

The authors have discussed the limitations in Section 6.

---

> ### Author Rebuttal · Authors · 2023-08-09
>
> Thank you for your supportive review and insightful suggestions. Please find below our specific responses.
>
> 1. Thank you for suggesting experiments on other architectures. We conducted a preliminary study on a lightweight BERT model which has 12 layers, 256 model dimensions, and 4 attention heads. On masked language pretraining task (trained on the BERT pretraining data), we observe lightweight-BERT achieves 54.7 MLM accuracy while lightweight-BERT + AltUpx2 achieves 56.2 MLM accuracy. We will include these empirical results in the updated version of the paper.
> 2. Thank you for pointing out this potential source of ambiguity and for your suggestion. You are fully right that Sequence-AltUp is different from the widened representations idea since it applies the predict-compute-correct idea of Algorithm 1 to the sequence dimension but does not increase the sequence dimension. We revised our paper to clarify the text surrounding Sequence-AltUp based on your helpful suggestion of framing it as increasing the effective width of each token.
> 3. You are right that Conformer is not an appropriate example for “striding operation” on Line 14. We have updated the paper to remove this ambiguity. Thank you for your careful consideration of our work.
> 4. That is a great point – we definitely believe that a larger number of train steps can help with the infrequent activations that arise when we use a large value of $K$. Our ongoing work is focused on training recipes to enable the application of AltUp with a very large expansion factor $K$.

---

> > ### Comment · Reviewer_raib · 2023-08-15
> >
> > Thank you for addressing my points. I will keep my score.

---

### Official Review · Reviewer_UGzF · 2023-07-06

**Soundness:** 3 good
**Presentation:** 3 good
**Contribution:** 2 fair
**Rating:** 7
**Confidence:** 3

**Summary:**


The study introduces Alternating Updates, a novel method to increase the capacity of transformer models without significantly raising latency. AltUp broadens the token representation, operating on a subblock of the widened representation at each layer, and employs a predict-and-correct mechanism for updating inactive blocks. AltUp also extends to the sequence dimension and can work synergistically with existing techniques like Sparse Mixture-of-Experts models. This allows for the creation of efficient models with high capacity. The effectiveness of the method is demonstrated across various scenarios, with notable performance improvements in language modeling and understanding benchmarks. It allows for a speedup of up to 87% on SuperGLUE and SQuAD benchmarks without accuracy loss.

**Strengths:**

- the paper is scientifically sound and well written

**Weaknesses:**

The authors did not mention other efficient transformer variants and their latency times (e.g. Flash attention), therefore it is hard to judge the model's performance.

**Questions:**

- how the model would perform on the LRA benchmark?
- how is it comparable with other efficient attention variations?
- does the model scales up to other domains beyond GLUE and SuperGLUE bechmarks?

**Limitations:**

The authors have openly discussed some limitations of their work, recognizing that there is a lack of a deep theoretical understanding of the properties of their proposed technique, AltUp, given the complexity of analyzing transformer models. They acknowledge that the optimal hyperparameter K might vary on an application-specific basis and plan to investigate this in the future.

---

> ### Author Rebuttal · Authors · 2023-08-09
>
> We are grateful for the reviewer’s careful consideration of our paper and their helpful feedback. Please see our specific comments below.
>
> ### Comparison to other efficient transformer variants
>
> As we mention in our response to Reviewer RDoa, the favorable properties of our method and its operation on the representation dimension make it difficult to directly compare to existing techniques which target different components of the architecture. For example, an MoE T5 model [1], Switch-Base, leads to a 2.5x speed-up measured in pretrain accuracy (not downstream evaluations) relative to a T5-Large dense model. However, Switch-Base contains 10x more parameters than a T5-Large dense model (7B vs. 0.7B), necessitates careful sharding along the model and expert dimensions, introduces auxiliary losses for load balancing and stability that need to be implemented, and requires tedious hyper-parameter tuning. Even when resources are available for sharding, there is a high degree of implementation and maintenance complexity involved with sharding-aware models. These challenges are in contrast to AltUp, which introduces an often negligible amount of additional parameters (especially with Recycled-AltUp), does not require sharding, and only introduces a single integral hyperparameter $K$ with a default value that works well across the scenarios we considered.
>
> More generally, we view AltUp as an orthogonal method that can work synergistically with existing approaches like MoE or Flash Attention (which provides up to 3x speedup [2]). We will include this contextualization in our final submission so that it is easier to judge the improvements with AltUp.
>
>
> ### Remaining questions
> 1. In our work, we adopted the standard T5 training procedure, with an input sequence length of 512 and target sequence length of 114.  This makes our trained models ill-suited for long input tasks, such as LRA. For example, LongT5 and CoLT5 models are T5 based models targeting long contexts and they are pre-trained with input sequence length of 4098 and target sequence length of 920. We leave long-context evaluations with our approach to future work.
> 2. As we highlight in our submission, our approach is a conditional computation approach where the conditionality is in activating a subblock of the token representation at each layer. So for AltUp with $K=2$, for example, the efficiency comes from using attention and MLP blocks that operate on a $d$-dimensional input, while we maintain a $2d$-dimensional representation from one layer to the next. This differs from the more well studied  efficient attention work that focuses on the *sequence* dimension with the goal of reducing the quadratic dependence of attention on the sequence length, e.g., Longformer, Linformer, Performer, BigBird. As with MoE, these efficient attention mechanisms are orthogonal to our approach and can be combined synergistically.
> 3. In addition to GLUE and SuperGLUE benchmarks, our evaluations (Fig. 4 and Table 1) contain results for SQuAD and TriviaQA. These datasets are considered fairly standard and comprehensive for the evaluation of T5 models [4]. Evaluations on translation tasks are not feasible due to the monolingual vocabulary. Nevertheless, we have recently obtained promising preliminary results on MBPP and MATH datasets that follow the same trends as those in our submission.
>
>
> [1] https://arxiv.org/abs/2101.03961
>
> [2] https://arxiv.org/abs/2205.14135
>
> [3] https://huggingface.co/blog/long-range-transformers
>
> [4] https://arxiv.org/pdf/1910.10683.pdf

---

> > ### Comment · Reviewer_UGzF · 2023-08-16
> >
> > Thank you for your comments, I've update the score to "accept"

---

### Official Review · Reviewer_RDoa · 2023-07-06

**Soundness:** 2 fair
**Presentation:** 3 good
**Contribution:** 3 good
**Rating:** 6
**Confidence:** 3

**Summary:**

This paper proposes AltUp, a new method for reducing the inference cost of Transformers by not computing blocks of the FFN layers. The authors find real-world speedups at inference time without sacrificing accuracy on benchmark tasks.

**Strengths:**

The results are strong - it is impressive to get real-world speedup using sparsity without sacrificing accuracy. The idea is simple and appears to work well on hardware. Overall good work.

**Weaknesses:**

The evaluation section is missing baselines to compare against. It is hard to evaluate novelty without also evaluating against similar works that use sparsity to accelerate model inference.

One example of recent work that is very similar is Deja Vu (https://openreview.net/forum?id=wIPIhHd00i). I'm not 100% sure when the paper went public relative to the NeurIPS deadline so it is fine if that one turns out to be concurrent, but it is a little odd to me that there would be no baselines at all.

It would also be useful to contextualize the results against complementary methods such as quantization.

**Questions:**

1. What are similar baselines in the literature that should be compared against to contextualize the strength and novelty of results?

**Limitations:**

Yes

---

> ### Author Rebuttal · Authors · 2023-08-09
>
> Thank you for your supportive review and helpful feedback.  Please see our specific comments below.
>
> ### Comparison to Deja Vu
>
> Thank you for your helpful reference to Deja Vu [1]. As we state in our coverage of prior work, virtually all prior approaches in conditional computation, such as MoE, apply to selecting a subset of parameters of the MLPs and/or attention blocks to activate, usually in an input-specific way. This includes the work in Deja Vu, which centers around selecting subsets of the attention and MLP parameters to apply for each input (contextual sparsity) . Our work is orthogonal to Deja Vu and synergistic with these approaches at large as it focuses on conditional computation along the *representation dimension*  (shown in Sec. C of the appendix for MoE, for example). However, as you guessed, we weren’t aware of Deja Vu at the time of our submission and will include it in a discussion of related work.
>
>
> ### Contextualization of the strength and novelty of results relative to baselines
>
> To the best of our knowledge, our work is the first in conditional **representation** computation for transformers, which makes it challenging to find similar baselines for comparison. We would like to emphasize that the dense T5 baselines depicted in Figures 4 and 5, as well as our comparisons to 2x and 4x dense baselines in Table 4 of the appendix, serve as comparison points since these models are well-optimized and popularly-deployed T5 models. Any speedups over these baseline dense models with Alternating Updates signify gains that can be further improved by combining it with other techniques such as Deja Vu, MoE, or quantization. For instance, Sec. C of the supplementary material depicts this synergistic combination with MoE (see Table 6).
>
>
> The favorable properties of our method and its operation on the **representation** dimension make it difficult to directly compare to existing techniques. For example, an MoE T5 model [2], Switch-Base, leads to a 2.5x speed-up measured in pretrain accuracy (not downstream evaluations) relative to a T5-Large dense model. However, Switch-Base contains 10x more parameters than a T5-Large dense model (7B vs. 0.7B), necessitates careful sharding along the model and expert dimensions, introduces auxiliary losses for load balancing and stability that need to be implemented, and requires tedious hyper-parameter tuning. Even when resources are available for sharding, there is a high degree of implementation and maintenance complexity involved with sharding-aware models. These challenges are in contrast to AltUp, which introduces an often negligible amount of additional parameters (especially with Recycled-AltUp), does not require sharding, and only introduces a single integral hyperparameter $K$ with a default value that works well across the scenarios we considered.
>
> More generally, we view AltUp as an orthogonal method that can work synergistically with existing approaches like MoE or quantization (which reportedly provides 4-5x speedup [3]). Thank you again for your constructive feedback, we will include this discussion surrounding the contextualization of our results and comparison to baselines in our final submission.
>
>
> [1] https://openreview.net/forum?id=wIPIhHd00i
>
> [2] https://arxiv.org/abs/2101.03961
>
> [3] https://proceedings.neurips.cc/paper_files/paper/2022/hash/adf7fa39d65e2983d724ff7da57f00ac-Abstract-Conference.html

---

> > ### Comment · Reviewer_RDoa · 2023-08-11
> >
> > Thank you for answering my questions, and I look forward to the updated camera ready. I will be keeping my score a 6.

---

### Decision · Program_Chairs · 2023-09-21

**Decision:**

Accept (spotlight)

**Comment:**

The paper proposes a new method to expand the hidden state size of neural network models. The proposed method is both computational and parameter-efficient. All reviewers agree that the paper is strong enough to be accepted by NeurIPS.